# Antiestrogenic Activity and Possible Mode of Action of Certain New Nonsteroidal Coumarin-4-acetamides

**DOI:** 10.3390/molecules25071553

**Published:** 2020-03-28

**Authors:** Maha S. Almutairi, Areej N. Al Suwayyid, Amal Aldarwesh, Omaima M. Aboulwafa, Mohamed I. Attia

**Affiliations:** 1Department of Pharmaceutical Chemistry, College of Pharmacy, King Saud University, Riyadh 11451, Saudi Arabia; asuwayyid@gmail.com; 2Saudi Food and Drug Authority, Riyadh 13513, Saudi Arabia; 3Department of Optometry and Visual Sciences, College of Applied Medical Sciences, King Saud University, Riyadh 12372, Saudi Arabia; aaldarweesh@ksu.edu.sa; 4Department of Pharmaceutical Chemistry, Faculty of Pharmacy, Alexandria University, Alexandria 21500, Egypt; omaimawafa@yahoo.com; 5Medicinal and Pharmaceutical Chemistry Department, Pharmaceutical and Drug Industries Research Division, National Research Centre (ID: 60014618), El Bohooth Street, Dokki, Giza 12622, Egypt

**Keywords:** coumarin-4-acetamide, nonsteroidal antiestrogens, SERMs, aromatase inhibitors

## Abstract

The preparation of certain 2-(2-oxo-2*H*-chromen-4-yl)-*N*-substituted acetamides **IIIa–h** was planned as a step in the development of new modified nonsteroidal antiestrogens. The purity of target compounds **IIIa–h** was checked by thin-layer chromatography (TLC), and their structures were confirmed using various spectroscopic tools including IR, ^1^H-NMR, ^13^C-NMR, and MS spectroscopy. Viability tests were applied using 3-(4,5-dimethylthiazol-2-yl)-2,5-diphenyl tetrazolium bromide (MTT) assay to evaluate the cytotoxic effect of the synthesized compounds against two breast cancer cell lines, MCF-7 and MDA-MB-231. Compound **IIIb** proved the most active against MCF-7 cells, with an IC_50_ value of 0.32 μM. The results of an analysis of in vitro antiestrogenic activity indicated that only compound **IIIb** exhibited antiestrogenic activity; its IC_50_ value of 29.49 μM was about twice as potent as that of the reference compound, MIBP. The aromatase activity was evaluated for the synthesized target compounds **IIIa–g** and the intermediates **Ib** and **IIa**. A significant aromatase inhibition was observed for the intermediate **Ib** and compound **IIIe**, with IC_50_ values of 14.5 and 17.4 μM, respectively. Compound **IIIb**, namely 7-methoxy-4-(2-oxo-2-(piperidin-1-yl)ethyl)-2H-chromen-2-one, could be used as an antiestrogen and/or cytotoxic agent with selective activity against tumor cells.

## 1. Introduction

Breast cancer is one of the most common and devastating cancers in women worldwide. According to researchers in the United States, the number of new cases of female breast cancer is 124.9 per 100,000 women per year [1]. In 2018, out of 5411 breast cancer cases diagnosed in Saudi Arabia, 708 patients (13.08%) died [2]. The first prognostic and predictive factor of breast cancer is related to estrogen receptors (ER) [3,4]. The development of ER antagonists has enabled successful treatment of postmenopausal women with hormone-dependent breast cancers [5]. Increased levels of estrogens are associated with tumor growth in endocrine-dependent tissues [6].

The coumarin (benzopyran-2-one, chromen-2-one,) ring system was used in research over 200 years ago. The name coumarin is derived from *Coumarouna odorata Aube*, from which it was isolated for the first time in 1820 [7]. Coumarins exhibit relatively low toxicity and participate in a remarkable array of biological activities [8,9,10,11,12,13]. One of the most interesting among the diverse biological activities of coumarins is their anti-breast cancer activity.

During the identification of potent coumarin-based selective estrogen receptor modulators (SERM) compounds, the goal was to find a potent estrogen inhibitor with an interleukin-6 (IL-6) inhibitory activity [14]. The presence of an amine side-chain in compound SP500263 (Figure 1) plays a key role in determining its SERM activity; however, this compound also showed undesirable agonist activity in the MCF-7 proliferation assay [15,16]. The core structures of SERMs are diverse, including triphenylethylene [17,18], benzothiophene, chromene (benzopyran) [19,20,21,22,23], naphthalene, indole, and steroid derivatives [24].

Unfortunately, anti-breast cancer drugs like tamoxifen (TAM) [18] achieve significant clinical results in only 30–40% of patients, because drug resistance usually develops after one or two years of treatment [25]. Among the fourth generation of SERMs, benzopyrans are potent antiestrogenic compounds with high oral bioavailability that have recently been tested in vitro and in vivo. These compounds display a 1.5–2.9-fold greater affinity than 17β-estradiol (E2) for the estrogen receptors in human breast cancer and normal uterine cytosol [19]. These compounds have no agonistic estrogenic activity in the in vitro human breast cancer models and in vivo in nude mice [19]. The nonsteroidal antiestrogen acolbifene (Figure 1) is the most potent antiestrogen in terms of inhibition of both ERα and ERβ [7,20,26]. This compound has a number of advantages over all other antiestrogens and should be investigated for the treatment of ER-positive breast cancer and other estrogen-sensitive malignancies [20,21]. In addition, aromatase enzyme is involved in the last step of the estrogen biosynthetic pathway [27]. A number of coumarin derivatives bearing an imidazole ring at position four have been designed and synthesized as strong and selective aromatase inhibitors (AIs) [28].

From the practical point of view, the compounds of interest contain substituted acetamide functionalities attached to the four position of coumarin nuclei, while maintaining all functionalities at their relative positions in the designed compounds. This pattern might dramatically affect the binding to the ER and subsequently the enhancement of their biological activity as anti-breast cancer agents.

## 2. Results and Discussion

### 2.1. Chemistry

Coumarin (2*H*-Chromen-2-one) has been synthesized by several methods, including Pechman–Duisberg condensation [29]. According to Scheme 1, the intermediates methyl (6-methoxy-2-oxo-2*H*-chromen-4-yl)acetate (**Ia**), methyl (7-methoxy-2-oxo-2*H*-chromen-4-yl)acetate (**Ib**), (6-methoxy-2-oxo-2*H*-chromen-4-yl)acetic acid (**IIa**), and (7-methoxy-2-oxo-2*H*-chromen-4-yl)acetic acid (**IIb**) were required for the synthesis of the target compounds **IIIa–h**. The reaction sequence to prepare the title compounds **IIIa–h** is outlined in Scheme 1.

The ^1^H-NMR spectra of compounds **Ia,b** exhibited a singlet peak at 3.77 ppm, which was assigned to methyl ester (COOCH_3_) protons, while the disappearance of the characteristic peak at 12.81 ppm for the carboxylic acid proton was observed. ^13^C-NMR spectroscopy indicated the presence of a new carbon peak at 52.68 ppm, which was assigned to be the methyl ester carbon (COOCH_3_).

The direct reaction of a carboxylic acid with an amine proceeds smoothly with aliphatic secondary amines like piperidine producing the target compounds **IIIa**,**b**. On the other hand, the best method for the preparation of compounds **IIIc**–**h** was the use of a 1-ethyl-3-[3-(dimethylamino)propyl] carbodiimide HCl (EDCI.HCl) reagent, which was added to a mixture of the appropriate aromatic amine and the acids **IIa**,**b** in dimethyformamide (DMF).

The structures of compounds **IIIa**–**h** were confirmed by their IR spectra, in which the appearance of C–N bands at 1026 cm^−1^ and 1024 cm^−1^ indicates compounds **IIIa** and **IIIb**, respectively. The disappearance of broad –OH bands suggested the formation of a new amide bond for compounds **IIIc–h**. The ^1^H-NMR spectra of compounds **IIIa**,**b** did not show –NH amide protons as they are tertiary amides. In addition, the ^1^H-NMR spectra of compounds **IIIc–h** were characterized by the appearance of –NH peaks that resonate at 10.08–10.53 ppm as singlets. Moreover, the ^13^C-NMR spectra of compounds **IIIc–h** were characterized by the presence of six new aromatic carbons in the range of 108.2–147.9 ppm. The ^13^C-NMR and DEPT spectra of the compound **IIIa** were well resolved and confirmed the presence of all carbon atoms in the molecule. On other hand, the HSQC spectrum of the compound **IIIa** specified its proton–carbon coupling.

### 2.2. Biological Evaluation

#### 2.2.1. In Vitro Cytotoxicity

The biological investigation started with the evaluation of the cytotoxic activity of the intermediates **Ia–b** and **IIa–b** (Table 1). Compounds **Ib** and **IIa** had a highly cytotoxic effect against the MCF-7 cell line, with IC_50_ values of 1.44 and 1.00 μM, respectively. However, only compound **Ib** was highly cytotoxic to the MDA-MB-231 cell line, with its IC_50_ value of 1.00 μM being about 19 times more potent than the reference standard camptothecin (IC_50_ value of 19.24 μM, Table 1 and Figure 2). Methyl (7-methoxy-2-oxo-2*H*-chromen-4-yl)acetate (**Ib**) was highly cytotoxic towards both breast cancer cell lines.

The cytotoxic activity of the synthesized compounds **IIIa–h** was evaluated in vitro against the human breast adenocarcinoma MCF7 (ER^+^ breast cancer cell line) and the MDA-MB-231 (triple-negative breast cancer cell line, TNBC), using camptothecin as a pyranone-bearing reference standard. Treatment of the ER+ MCF-7 and MDA-MB-231 cell lines with 0.39, 1.56, 6.25, 25, and 100 μM concentrations of the synthesized target compounds **IIIa–h** resulted in a significant cytotoxic effect. The most active compounds against the MCF-7 breast cancer cell line were subjected to in vitro antiestrogenic activity and in vitro aromatase inhibition activity.

Considering the cytotoxicity of *N*-substituted coumarin-4-acetamides **IIIa–h** (Table 2 and Figure 3), compound **IIIb** is most active against MCF-7 cells with an IC_50_ value of 0.32 μM, as compared with the reference cytotoxic compound, camptothecin (IC_50_ = 4.41 μM). In addition, compound **IIIe** is the most active candidate against the MDA-MB-231 cell line with an IC_50_ value of 2.14 μM, while camptothecin displayed an IC_50_ value of 19.24 μM against the same cell line.

In general, 7-methoxycoumarin-4-acetamide derivatives **IIIb** and **IIIf–h** showed a greater in vitro cytotoxic effect against MCF-7 cells as compared with their positional isomers 6-methoxycoumarin derivatives **IIIa** and **IIIc–e**. The most active compounds are in the following order: **IIIb** > **IIIg** > **IIIa** > **IIId**, with IC_50_ values of 0.32, 0.72, 1.82, and 2.80 μM, respectively. On the other hand, the results of the cytotoxic evaluation of the acetamides **IIIa–h** against MDA-MB-231 human breast cancer cells indicated that compounds **IIIc**, **e**, and **f** had a highly cytotoxic effect, with IC_50_ values of 4.6, 2.1, and 6.9 μM, respectively (Table 2 and Figure 3).

#### 2.2.2. In Vitro Antiestrogenic Activity

The in vitro antiestrogenic activity of the selected target compounds **IIIa**,**b**, **IIId**, and **IIIf**,**g** was studied through an estrogen-dependent human breast cancer MCF-7 cell proliferation assay in the presence of 17-*β*-estradiol. The ability of the tested compounds to inhibit cell proliferation induced by 17-*β*-estradiol was determined (Table 3 and Figure 4). One of the disadvantages of the compound SP500263 was that it had an undesirable agonist activity in the MCF-7 proliferation assay. However, among the synthesized compounds, compound **IIIb** is the most active, with its IC_50_ value of 29.49 μM being about twice as potent as the reference compound, MIBP.

#### 2.2.3. In Vitro Aromatase Inhibition

One of the most successful targeted breast cancer therapies is the inhibition of the AR enzyme, which catalyzes the rate-limiting final step of estrogen biosynthesis by modulating ER. To ensure cytotoxicity against the MCF-7 breast cancer cell line, aromatase enzymatic activity was assayed using the aromatase inhibitor (AI), letrozole, as a reference standard. Compound **Ib** was equipotent to letrozole (IC_50_ = 15.03 μM) aromatase inhibitory activity with an IC_50_ value of 14.5 μM (Table 4 and Figure 5).

Moreover, acetamides **IIIa–g** showed low or no aromatase inhibition activity (Table 4 and Figure 5), except compound **IIIe**, which showed moderate inhibitory activity against aromatase (IC_50_ = 17.38 μM), alongside its promising activity towards the MDA-MB-231 human breast cancer cell line (IC_50_ value of 2.14 μM).

## 3. Experimental

### 3.1. Chemistry

#### 3.1.1. General

Melting points were determined in open glass capillaries on an electrothermal melting point apparatus and are uncorrected. Infrared (IR) spectra were recorded for potassium bromide discs ν (cm^−1^) on an IR affinity-1s Fourier transform infrared spectrophotometer. The ^1^H-NMR and ^13^C-NMR spectra were determined on a Bruker (700 MHz) (Coventry, Germany) and Agilent Technologies (600 MHz) (Palo Alto, CA, USA) spectrometer. Correlation spectroscopy: ^1^H, ^13^C, HSQC, and DEPT spectra were recorded on a Bruker (700 MHz) spectrometer. Chemical shifts are expressed as δ values (ppm), using tetramethylsilane (TMS) as an internal reference. Signals are indicated by the following abbreviations: s = singlet, d = doublet, t = triplet, q = quartet, m = multiplet, br = broad. Mass spectra (MS) were obtained on a GCMS QP 2010 Ultra/SE (Ver. 4.20), GCMS-TQ Series (Ver. 4.20) Shimadzu apparatus (Kyoto, Japan). Elemental analyses were carried out at Research Center, College of Pharmacy, King Saud University, Saudi Arabia, and the results agreed favorably with the proposed structures within ± 0.4% of the theoretical values. Follow-up of the reaction and checking on the homogeneity of the compounds were made by performing thin-layer chromatography (TLC) on pre-coated (0.25 mm) (GF 254) silica gel plates. Routinely used developing solvents were volume to volume: C_6_H_14_:EtOAc (6:4); C_6_H_14_:EtOAc (5:5); MeOH:CHCl_3_ (1:9). Visualization of the spots was performed by exposure to a UV lamp at 254 nm or to iodine vapors. All statistical analyses were carried out using GraphPad Prism (San Diego, CA, US) version 6.0 software. Statistical analysis was conducted using one-way ANOVA, followed by multiple Tukey–Kramer post hoc tests at *p* < 0.05, which was considered a marker of statistical significance.

#### 3.1.2. General Procedure for the Synthesis of Methyl 2-(2-oxo-2*H*-chromen-4-yl)acetates **Ia,b**

Route A: To a stirred solution of dimethyl-acetone-1,3-dicarboxylate (1.83 mL, 2.2 g, 10 mmol), the appropriate portions of methoxyphenol (1.24 g, 10 mmol) and concentrated H_2_SO_4_ (1.12 mL, 1 g, 11 mmol) were added, each in three equal portions, in a cold environment (temperature not exceeding 10 °C). The resulting reaction mixture was stored at 0 °C overnight, then poured into ice-cold water (40 mL). The white precipitate that formed was filtered off and re-crystallized from methanol to give the respective methyl 2-(2-oxo-2*H*-chromen-4-yl)acetate derivatives **Ia**,**b** [30].

Route B: A few drops of H_2_SO_4_ were added to a solution of 2-(7-methoxy-2-oxo-2*H*-chromen-4-yl)acetic acid (**IIb**, 1 mg, 4 mmol) in MeOH (10 mL). The resulting solution was refluxed for 5 h. After completion of the reaction, as indicated by TLC (ethylacetate/hexane, 4/6), MeOH was evaporated under reduced pressure. The residue was dissolved in ethyl acetate (20 mL), washed with NaHCO_3_ (20 mL), dried over anhydrous Na_2_SO_4_, and filtered. Ethyl acetate was removed in vacuo to produce compound **Ib** [31].

Methyl (6-methoxy-2-oxo-2*H*-chromen-4-yl)acetate (**Ia**):

Yield (route A): 0.5 g (20%); White glittery crystals m.p.: 153–154 °C; IR (KBr): ν (cm^−1^): 1724 (C=O, lactone), 1734 (C=O, ester), 2941 (C–H, aliphatic), 3088 (C–H, aromatic); ^1^H-NMR (DMSO-*d_6_*) δ ppm (600 MHz): 3.62 (s, 3H, ester CH_3_), 3.77 (s, 3H, Ar OCH_3_), 4.01 (s, 2H, CH_2_CO), 6.48 (s, 1H, H-3), 7.11 (s, 1H, H-5), 7.20 (s, 1H, H-7), 7.34 (s, 1H, H-8); ^13^C-NMR (DMSO-*d_6_*) δ ppm (150 MHz): 36.9 (CH_2_CO), 52.7 (ester CH_3_), 56.2 (OCH_3_), 108.8 (C-5), 117.3 (C-3), 118.2 (C-7), 119.5 (C-4a), 119.7 (C-8), 147.8 (C-8a), 149.5 (C-4), 155.9 (C-6), 160.2 (C-2), 170.1 (C=O); MS *m/z* (% relative abundance): [M]^+^ 248 (3.82).

Methyl (7-methoxy-2-oxo-2*H*-chromen-4-yl)acetate (**Ib**):

Yield by route A: 1.2 g (48%) and by route B: 0.7 g (70%); White crystals m.p.: 116–118 °C; IR (KBr): ν (cm^−1^):1724 (C=O, lactone), 1734 (C=O, ester), 2966 (C–H, aliphatic), 3093 (C–H, aromatic); ^1^H-NMR (DMSO-*d_6_*): δ ppm (600 MHz) 3.62 (s, 3H, ester CH_3_), 3.84 (s, 3H, ArOCH_3_), 3.97 (s, 2H, CH_2_CO), 6.30 (s, 1H, H-3), 6.94 (s, 1H, H-6), 7.00 (s, 1H, H-8) and 7.87 (s, 1H, H-5); ^13^C-NMR (DMSO-*d_6_*): δ ppm (150 MHz) 37.1 (CH_2_CO), 52.7 (ester CH_3_), 56.4 (OCH_3_), 101.4 (C-8), 112.8 (C-6), 113.5 (C-3), 114.8 (C-4a), 127.0 (C-5), 149.9 (C-8a), 155.4 (C-4), 160.5 (C-7), 162.9 (C-2), 170.1 (C=O); MS *m/z* (% relative abundance): [M]^+^ 248 (0.33).

#### 3.1.3. General Procedure for the Synthesis of Coumarin-4-Acetic acid Derivatives **IIa,b**

Route A: A mixture of citric acid monohydrate (4.2 g, 20 mmol) and concentrated H_2_SO_4_ (5.6 mL) was stirred at room temperature for 60 min, then slowly heated (rate of heating governed by foaming) to 70 °C. After 35 min at this temperature, with stirring throughout, the evolution of carbon monoxide had slackened, and the clear solution was rapidly cooled to 0 °C. Then the appropriate methoxyphenol (2 g, 16.1 mmol) and concentrated H_2_SO_4_ (2.24 mL) were added, each in three equal portions, to the stirred solution at such a rate that the internal temperature did not exceed 10 °C. The resulting reaction mixture was stored at 0 °C for 16 h, poured into ice cold water (40 mL), and the resulting crystalline precipitate filtered off and washed thoroughly with H_2_O. The collected solid was dissolved under stirring in 1N Na_2_CO_3_ solution (20 mL), heated for 15 min at 65 °C, and the insoluble matter was filtered off and washed with water (2 × 10 mL). The combined filtrate and washings were acidified with concentrated HCl to give the respective coumarin-4-acetic acid derivatives **IIa**,**b** [32].

Route B: A solution of methyl (6-methoxy-2-oxo-2*H*-chromen-4-yl)acetate (**Ia**, 1 g, 4 mmol) in ethanol (10 mL) and 0.5% NaOH (100 mL) was refluxed for 2 h. It was then cooled to room temperature, acidified with concentrated HCl to pH = 2, and cooled to 0 °C. The precipitated solid was filtered off, washed thoroughly with ethanol, and dried to give compound **IIa**, which was used for the next step without further purification [33,34].

(6-Methoxy-2-oxo-2*H*-chromen-4-yl)acetic acid (**IIa**):

Yield by route A: 0.23 g (6.1%) and by route B: 0.84 g (89%); light brown fluffy powder m.p.: 178–179 °C; IR (KBr): ν (cm^−1^): 1174–1255 (C–O), 1670 (C=O, carboxylic acid), 1724 (C=O, lactone), 2924 (C–H, aliphatic), 3088 (C–H, aromatic), 3200(OH); ^1^H-NMR (DMSO-*d_6_*): δ ppm (600 MHz) 3.78 (s, 3H, ArOCH_3_), 3.92 (s, 2H, CH_2_CO), 6.48 (s, 1H, H-3), 7.15 (s, 1H, H-5), 7.22 (s, 1H, H-7), 7.35 (s, 1H, H-8), 12.81 (s, 1H, COOH); ^13^C-NMR (DMSO-*d_6_*): δ ppm (150 MHz) 36.9 (CH_2_CO), 56.2 (OCH_3_), 109.0 (C-5), 114.9 (C-3), 117.1 (C-7), 118.1 (C-4a), 119.4 (C-8), 147.8 (C-8a), 150.2 (C-4), 155.9 (C-6), 160.2 (C-2), 171.1 (C=O); MS *m/z* (% relative abundance): [M]^+^ 234 (0.58), [M + 1] 235 (12.06).

7-Methoxy-2-oxo-2*H*-chromen-4-yl)acetic acid (**IIb**):

Yield (route A): 3.2 g (85%); White crystals m.p.: 186–187 °C (literature: 186 °C [35]; IR (KBr): ν (cm^−1^): 1664 (C=O, carboxylic acid), 1718 (C=O, lactone), 2926 (C–H, aliphatic), 3439 (br, OH); ^1^H-NMR (DMSO-*d_6_*): δ ppm (600 MHz) 3.82 (s, 5H, CH_2_CO &ArOCH_3_), 6.27 (s, 1H, H-3), 6.93 (s, 1H, H-6), 6.97 (s, 1H, H-8), 7.58 (s, 1H, H-5), 12.81 (s, 1H, COOH); ^13^C-NMR (DMSO-*d_6_*): δ ppm (150 MHz) 37.7 (CH_2_CO), 56.3 (OCH_3_), 101.3 (C-8), 112.7 (C-6), 112.9 (C-3), 113.3 (C-4a), 127.0 (C-5), 150.6 (C-8a), 155.4 (C-4), 160.5 (C-7), 162.8 (C-2), 171.1 (C=O); MS *m/z* (% relative abundance): [M]^+^ 234 (1.89), [M + 1] 235 (4.09).

#### 3.1.4. General Procedure for the Synthesis of 4-(2-oxo-2-(piperidin-1-yl)ethyl)-2*H*-chromen-2-one derivatives **IIIa,b**

The appropriate methyl (2-oxo-2*H*-chromen-4-yl)acetate **Ia**,**b** (0.248 g, 1 mmol) and piperidine (1 mL, 0.85 g, 1 mmol) was heated to reflux in toluene (10 mL) in the presence of a catalytic amount of *p*-toluene sulfonic acid (*p*-TSOH) for 4 h. The solvent was concentrated under reduced pressure and the precipitated solid was filtered off, washed with toluene, and dried to yield the respective target compounds **IIIa**,**b** [36].

6-Methoxy-4-(2-oxo-2-(piperidin-1-yl)ethyl)-2*H*-chromen-2-one (**IIIa**):

Yield: 0.21 (70%); Fluffy pale yellow powder m.p.: 167–168 °C; IR (KBr): ν (cm^−1^): 1026 (C–N), 1718 (C=O, amide), 1734 (C=O, lactone), 2939 (C–H, aliphatic), 3020 (C–H, aromatic); ^1^H-NMR (DMSO-*d_6_*): δ ppm (700 MHz) 1.46 (br s, 2H, CH_2_ piperdine), 1.57 (br s, 2H, CH_2_-piperdine), 1.62 (br s, 2H, CH_2_-piperdine), 3.47 (br s, 2H, CH_2_-piperdine), 3.53 (br s, 2H, CH_2_-piperdine), 3.81 (s, 3H, OCH_3_),4.03 (s, 2H, CH_2_), 6.38 (s, 1H, H-3), 7.11 (s, 1H, H-5), 7.24 (d, *J* = 9.1 Hz, 1H, H-7), 7.39 (d, *J* = 9.1 Hz, 1H, H-8); ^13^C-NMR (DMSO-*d_6_*): δ ppm (175 MHz) 24.3 (CH_2-_piperdine), 25.9 (CH_2-_piperdine), 26.7 (CH_2_ piperdine), 36.7 (CH_2_CO), 42.5 (CH_2_-piperdine), 46.9 (CH_2_-piperdine), 56.2 (OCH_3_), 109.4 (C-5), 116.5 (C-3), 117.9 (C-7), 119.1 (C-8), 120.3 (C-4a), 147.7 (C-8a), 152.1 (C-4), 155.8 (C-6), 160.3 (C-2), 166.7 (C=O); MS *m/z* (% relative abundance): [M]^+^ 301 (0.48).

7-Methoxy-4-(2-oxo-2-(piperidin-1-yl)ethyl)-2*H*-chromen-2-one (**IIIb**):

Yield: 0.21 g (70%); White crystals m.p.: 135–136 °C; IR (KBr): ν (cm^−1^): 1024 (C–N), 1718 (C=O, amide), 1734 (C=O, lactone), 2949 (C–H, aliphatic), 3095 (C–H, aromatic); ^1^H-NMR (DMSO-*d_6_*) δ ppm (700 MHz):1.44–1.47 (m, 2H, CH_2_-piperdine), 1.56–1.57 (m, 2H, CH_2_-piperdine),1.61–1.63 (m, 2H, CH_2_-piperdine), 3.45 (t, *J* = 5.6 Hz, 2H, CH_2_-piperdine), 3.51 (t, *J* = 5.6 Hz, 2H, CH_2_-piperdine), 3.87 (s, 3H, OCH_3_), 3.98 (s, 2H, CH_2_), 6.19 (s, 1H, H-3), 6.96 (dd, *J* = 9.1, 2.8 Hz, 1H, H-6), 7.01(d, *J* = 2.8 Hz, 1H, H-8), 7.57 (d, *J* = 9.1 Hz, 1H, H-5); ^13^C-NMR (DMSO-*d_6_*): δ ppm (175 MHz) 24.4 (CH_2_-piperdine), 25.8 (CH_2_-piperdine), 26.5 (CH_2_-piperdine), 31.2 (CH_2_CO), 42.7 (CH_2_-piperdine), 46.8 (CH_2_-piperdine), 56.4 (OCH_3_), 101.1 (C-8), 112.6 (C-6), 125.9 (C-3), 127.3 (C-4a), 128.5 (C-5), 152.6 (C-8a), 155.3 (C-4), 160.6 (C-7), 162.7 (C-2), 166.7 (C=O); MS *m/z* (% relative abundance): [M]^+^ 301 (10.33).

#### 3.1.5. General Procedure for the Synthesis of 2-(2-oxo-2*H*-chromen-4-yl)-*N*-phenylacetamides **IIIc-h**

1-Ethyl-3-[3-(dimethylamino)propyl]carbodiimide HCl (EDCI.HCl, 0.47 g, 3 mmol) was slowly added to a stirred solution containing the appropriate 2-oxo-2*H*-chromen-4-yl)acetic acid derivative **IIa**,**b** (0.234 g, 1 mmol) and the proper arylamine (1.2 mmol) in DMF (15 mL) at 0 °C. After 2 h, the reaction mixture was warmed to room temperature and stirring was continued for an additional 24 h. Thereafter, the reaction mixture was poured into H_2_O (22 mL). The separated solid was filtered off under suctioning, washed repeatedly with H_2_O, and dried to give compounds **IIIc–h**.

2-(6-Methoxy-2-oxo-2*H*-chromen-4-yl)-*N*-phenylacetamide (**IIIc**):

Yield: 0.27 g (87%); White powder m.p.: 219–220 °C; IR (KBr): ν (cm^−1^) 1043 (C–N), 1078–1213 (C–O), 1660 (C=O, amide), 1734 (C=O, lactone), 2960 (C–H, aliphatic), 3059 (C–H, aromatic), 3286 (NH, secondary amine); ^1^H-NMR (DMSO-*d_6_*): δ ppm (700 MHz) 3.80 (s, 3H, OCH_3_), 3.99 (s, 2H, CH_2_), 6.54 (s, 1H, H-3), 7.08 (t, *J* = 7.7 Hz, 1H, H-4′), 7.26 (dd, *J* = 9.1, 2.8 Hz, 1H, H-7), 7.32–7.34 (m, 3H, H-5 and H-3′ & H-5′), 7.40 (d, *J* = 8.4 Hz, 1H, H-8), 7.71 (d, *J* = 8.4 Hz, 2H, H-2′ & H-6″), 10.42 (s, 1H, NH); ^13^C-NMR (DMSO-*d_6_*): δ ppm (175 MHz) 31.3 (CH_2_CO), 56.3 (OCH_3_), 108.8 (C-5), 117.2 (C-3), 118.2 (C-7), 119.4 (C-4a), 119.7 (C-2′ & C-6′), 120.1 (C-8), 124.1 (C-4′), 129.3 (C-3′ &C-5′), 139.2 (C-1′), 147.9 (C-8a), 150.8 (C-4), 156.0 (C-6), 160.3 (C-2), 167.0 (C=O); MS *m/z* (% relative abundance): [M]^+^ 309 (0.73).

*N*-(4-Bromophenyl)-2-(6-methoxy-2-oxo-2*H*-chromen-4-yl)acetamide (**IIId**):

Yield: 0.19 g (48%); Silver powder m.p.: 202–203 °C; IR (KBr): ν (cm^−1^) 600 (C-Br), 1031 (C–N), 1070–1246 (C–O), 1653 (C=O, amide), 1730 (C=O, lactone), 2978 (C–H, aliphatic), 3088 (C–H, aromatic), 3228 (NH, secondary amine); ^1^H-NMR (DMSO-*d_6_*): δ ppm (700 MHz) 3.82 (s, 3H, OCH_3_), 4.05 (d, *J* = 7.0 Hz, 2H, CH_2_), 6.53 (s, 1H, H-3), 7.16 (d, *J* = 2.8 Hz, 1H, H-5), 7.26 (dd, *J* = 9.1, 2.8 Hz, 1H, H-7), 7.39 (d, *J* = 9.1 Hz, 1H, H-8), 7.51 (d, *J* = 9.1 Hz, 2H, H-2′ &H-6″), 7.56 (d, *J* = 9.1 Hz, 2H, H-3′ & H-5′), 10.53 (s, 1H, NH); ^13^C-NMR (DMSO-*d_6_*): δ ppm (175 MHz) 31.2 (CH_2_CO), 56.2 (OCH_3_), 108.8 (C-5), 115.7 (C-3), 117.3 (C-7), 118.2 (C-2′ &C-6′), 119.4 (C-4a), 119.7 (C-4′), 121.7 (C-8), 132.2 (C-3′ &C-5′), 138.6 (C-1′), 147.9 (C-8a), 149.7 (C-4), 156.0 (C-6), 160.2 (C-2), 169.6 (C=O); MS *m/z* (% relative abundance): [M]^+^ 388 (0.48).

N-(3-Hydroxy-4-methoxyphenyl)-2-(6-methoxy-2-oxo-2*H*-chromen-4-yl)acetamide (**IIIe**):

Yield: 0.21 g (59%); Brown powder m.p.: 173–174 °C; IR (KBr): ν (cm^−1^): 1028 (C–N), 1043–1257 (C–O), 1685 (C=O, amide), 1734 (C=O, lactone), 2935 (C–H, aliphatic), 3083 (C–H, aromatic), 3253 (NH, secondary amine), 2446–3527 (OH); ^1^H-NMR (DMSO-*d_6_*): δ ppm (700 MHz) 3.72 (s, 3H, C-4′-OCH_3_), 3.80 (s, 3H, OCH_3_), 3.92 (s, 2H, CH_2_), 6.51 (s, 1H, H-3), 6.84 (d, *J* = 8.4 Hz, 1H, H-5′), 6.93 (dd, *J* = 8.4, 2.8 Hz, 1H, H-6′), 7.13 (d, *J* = 2.1 Hz, 1H, H-2′), 7.25 (dd, *J* = 9.1, 2.8 Hz, 1H, H-7), 7.32 (d, *J* = 2.8 Hz, 1H, H-5), 7.40 (d, *J* = 9.1 Hz, 1H, H-8), 9.10 (s, 1H, OH), 10.15 (s, 1H, NH); ^13^C-NMR (DMSO-*d_6_*): δ ppm (175 MHz) 31.2 (CH_2_CO), 56.3 (OCH_3_), 108.3 (C-2′), 108.9 (C-5), 110.5 (C-3), 112.9 (C-5′), 117.1 (C-6′), 118.2 (C-7), 119.4 (C-4a), 120.1 (C-8), 132.7 (C-1′), 144.5 (C-3′), 146.9 (C-8a), 147.8 (C-4′), 151.0 (C-4), 156.0 (C-6), 160.3 (C-2), 166.4 (C=O); MS *m/z* (% relative abundance): [M]^+^ 355 (0.59), [M + 1] 356 (35.71).

2-(7-Methoxy-2-oxo-2*H*-chromen-4-yl)-*N*-phenylacetamide (**IIIf**):

Yield: 0.2 g (65%); Yellow-brown powder m.p.: 217–218 °C; IR (KBr): ν (cm^−1^): 1028 (C–N), 1647 (C=O, amide), 1728 (C=O, lactone), 2978 (C–H, aliphatic), 3074 (C–H, aromatic), 3265 (NH, secondary amine); ^1^H-NMR (DMSO-*d_6_*): δ ppm (700 MHz) 3.87 (s, 3H, OCH_3_), 3.95 (s, 2H, CH_2_), 6.34 (s, 1H, H-3), 7.00 (dd, *J* = 9.1, 2.8 Hz, 1H, H-6), 7.03 (d, *J* = 2.8 Hz, 1H, H-8), 7.07 (t, *J* = 7.7 Hz, 1H, H-4′), 7.32 (t, *J* = 7.7 Hz, 2H, H-3′ & H-5′), 7.58 (d, *J* = 7.7 Hz, 2H, H-2′& H-6′), 7.75 (d, *J* = 8.4 Hz, 1H, H-5), 10.35 (s, 1H, NH); ^13^C-NMR (DMSO-*d_6_*): δ ppm (175 MHz) 39.6 (CH_2_CO), 56.4 (OCH_3_), 101.4 (C-8), 112.7 (C-6), 113.1 (C-3), 113.4 (C-4a), 119.7 (C-2′ & C-6′), 124.1 (C-5), 127.0 (C-4′), 129.3 (C-3′ & C-5′), 139.3 (C-1′), 151.3 (C-8a), 155.4 (C-4), 160.5 (C-7), 162.9 (C-2), 167.0 (C=O); MS *m/z* (% relative abundance): [M + 1] 310 (33).

*N*-(4-Bromophenyl)-2-(7-methoxy-2-oxo-2*H*-chromen-4-yl)acetamide (**IIIg**):

Yield: 0.26 g (51%); Beige powder m.p.: 212–213 °C; IR (KBr): ν (cm^−1^): 500 (C-Br), 1066 (C–N), 1653 (C=O, amide), 1728 (C=O, lactone), 2935 (C–H, aliphatic), 3111 (C–H, aromatic), 3248 (NH, secondary amine); ^1^H-NMR (DMSO-*d_6_*): δ ppm (700 MHz) 3.87 (s, 3H, OCH_3_), 3.95 (s, 2H, CH_2_), 6.34 (s, 1H, H-3), 7.00 (dd, *J* = 8.4, 2.8 Hz, 1H, H-6), 7.04 (d, *J* = 2.8 Hz, 1H, H-8), 7.51 (d, *J* = 9.1 Hz, 2H, H-2′ & H-6′), 7.55 (d, *J* = 9.1 Hz, 2H, H-3′ & H-5′), 7.73 (d, *J* = 9.1 Hz, 1H, H-5), 10.48 (s, 1H, NH); ^13^C-NMR (DMSO-*d_6_*): δ ppm (175 MHz) 31.2 (CH_2_CO), 56.4 (OCH_3_), 101.4 (C-8), 112.7 (C-6), 113.1 (C-3), 113.5 (C-4a), 115.6 (C-2′ & C-6′), 121.6 (C-4′), 127.0 (C-5), 132.1 (C-3′ & C-5′), 139.6 (C-1′), 151.1 (C-8a), 155.4 (C-4), 160.5 (C-7), 162.9 (C-2), 167.3 (C=O); MS *m/z* (% relative abundance): [M]^+^ 388 (1.39), [M + 1] 389 (2.09), [M+2] 390 (1.29).

*N*-(3-Hydroxy-4-methoxyphenyl)-2-(7-methoxy-2-oxo-2*H*-chromen-4-yl)acetamide (**IIIh**):

Yield: 0.26 g (73%); Brown powder m.p.: 188–189 °C; IR (KBr): ν (cm^−1^): 1028 (C–N), 1653 (C=O, amide), 1707 (C=O, lactone), 2985 (C–H, aliphatic), 3100 (C–H, aromatic), 3228 (NH, secondary amine), 3327 (br OH); ^1^H-NMR (DMSO-*d_6_*): δ ppm (700 MHz) 3.72 (s, 3H, C-4′-OCH_3_), 3.88 (d, *J* = 7.0 Hz, 5H, OCH_3_ & CH_2_), 6.31 (s, 1H, H-3), 6.84 (d, *J* = 8.4 Hz, 1H, H-5′), 6.94 (dd, *J* = 9.1, 2.8 Hz, 1H, H-6′), 7.00 (dd, *J* = 9.1, 2.8 Hz, 1H, H-6), 7.02 (d, *J* = 2.8 Hz, 1H, H-8), 7.12 (d, *J* = 2.8 Hz, 1H, H-2′), 7.74 (d, *J* = 9.1 Hz, 1H, H-5), 9.09 (s, 1H, OH), 10.08 (s, 1H, NH); ^13^C-NMR (DMSO-*d_6_*): δ ppm (175 MHz) 31.2 (CH_2_CO), 56.3 (2 × OCH_3_), 101.4 (C-8), 108.2 (C-2′), 110.4 (C-6), 112.7 (C-3), 112.9 (C-5′), 113.1 (C-4a), 113.3 (C-6′), 127.0 (C-5), 132.8 (C-1′), 144.5 (C-3′), 146.9 (C-4′), 151.5 (C-8a), 155.4 (C-4), 160.6 (C-7), 162.9 (C-2), 166.4 (C=O); MS *m/z* (% relative abundance): [M + 2] 357 (27.25).

### 3.2. Biological Evaluation

#### 3.2.1. Cytotoxicity Assay (MTT Assay)

The cytotoxicity of compounds **Ia,b**, **IIa,b**, and **IIIa–h** against the MCF-7 cell line (ER+ breast cancer cell line) and MDA-MB-231 (triple-negative breast cancer cell line, TNBC) was determined using camptothecin as a pyranone-bearing reference standard [37]. The detailed experimental procedures are provided in the Appendix A.

#### 3.2.2. Antiestrogenic Activity

The antiestrogenic activity of the test compounds was examined by performing a 3-(4,5-dimethylthiazol-2-yl)-2,5-diphenyl tetrazolium bromide (MTT) assay of the MCF-7 cell line. In this experiment, MCF-7 cells were treated with 17β-estradiol (+ve cell proliferation compound). The effect of various concentrations of the tested compounds on cell proliferation in the presence of 17β-estradiol was measured [37]. The detailed experimental procedures are provided in the Appendix A.

#### 3.2.3. Aromatase Inhibition

Sandwich enzyme immunoassay was adopted for aromatase inhibition assessment [38]. The detailed experimental procedures are provided in the Appendix A.

## 4. Conclusions

In conclusion, 2-(2-oxo-2*H*-chromen-4-yl)-*N*-substituted acetamide derivatives **IIIa–h** have been prepared, characterized, and tested for their in vitro cytotoxic and antiestrogenic, as well as aromatase inhibition, activities. The target compounds **IIIa–h** showed variable cytotoxic activity against two breast cancer cell lines, MCF-7 and MDA-MB-231. 7-methoxy-4-(2-oxo-2-(piperidin-1-yl)ethyl)-2*H*-chromen-2-one (**IIIb**) was the most potent cytotoxic compound against MCF-7, being about 14-fold more potent than the reference standard, camptothecin. It also manifested high in vitro antiestrogenic activity (IC_50_ = 29.49 μM). These findings indicate that a cyclic aliphatic lipophilic substitution in compound **IIIb** produced obvious antiestrogenic and cytotoxic activities. Thus, it might have high affinity to ER. Unfortunately, the tested compounds show moderate to low aromatase inhibition activity, except for compound **IIIe**, which showed moderate inhibitory activity against aromatase with an IC_50_ value of 17.38 μM.

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
