# Peer review of "Antiestrogenic Activity and Possible Mode of Action of Certain New Nonsteroidal Coumarin-4-acetamides"

_molecules, 2020, doi:10.3390/molecules25071553_

Round 1

Reviewer 1 Report

The authors prepared a manuscript dealing with the synthesis of new non-steroidal coumarin-4-acetamides. They evaluated cytotoxic, antiestrogenic, and aromatase inhibition activities of the new coumarins and compared them with reference compounds. The manuscript is written in standard form and the procedures as well as results are clearly presented. However, before formulating aims of the work and conclusions, the authors should add a discussion about similar structures that have been already synthesized and studied in this field (in the Introduction section), and, compare biological data of their compounds with those of similar previously published structures (in the Results and discussion section). In this way, the comparison would be more complex, including not only reference substances but also some interesting recent molecules. In my opinion, the manuscript can be published in Molecules after this medium revision.

Reviewer 2 Report

In this manuscript the authors describe the synthesis new coumarin-4-acetic acid derivatives, their cytotoxic effects against two breast cancer cell lines (MCF-7 and MDA-MB-231) as well as their in vitro antiestrogenic and aromatase inhibitory activities.

Overall, the manuscript is well written and presented. The chemical synthesis is simple and the compounds are well characterized. But, to my opinion, their purity need to be confirmed (see my comments below). The cytotoxic and antiestrogenic inhibitory activities of selected synthesized compounds are interesting. On the other hand, the vast majority of the tested molecules did not exhibit aromatase inhibitory activity.

I have some suggestions and corrections that may improve the quality of the manuscript.

  1. The authors note (line 19-21) that “The purity of the target compounds IIIa-h was checked by thin layer chromatography (TLC) and their structures were confirmed using various spectroscopic tools including IR, 1H NMR, 13C NMR and MS spectroscopy.”

TLC cannot be considered sufficient analytical method for the determination of the purity of the synthesized compounds. The authors must provide either elemental analysis or chromatographic (e.g. HPLC) data (preferably accompanied by HRMS analysis data) as proof of purity of the synthesized compounds. 1H- and 13C-NMR spectra copies must be included in the Supplementary Material. Rf values must be included in the experimental part of the manuscript.

  1. Basically, the authors tried to investigate the cytotoxic, antiestrogenic and aromatase inhibitory potential of some 6-OCH3 or 7-OCH3 substituted coumarin-4-acetic acid derivatives, which are structurally related to known estrogen receptor modulator SP500263 and the non-steroidal antiestrogen acolbifene. Both of these known compounds bear a 7-hydroxy substituent. Did the authors consider the design and synthesis of 6 or 7-hydroxy substituted target molecules? If not, why?
  2. Line 23: Please, change “Compound IIIb being…” to “Compound IIIb proved…”
  3. Line 26: Please, change “value 29.49 μM” to “value of 9.49 μM”.
  4. Line 39: Please, change “…related to ER [3, 4]. The development of estrogen receptor (ER)..” to “…related to estrogen receptor (ER) [3, 4]. The development of ER…”.
  5. Line 41: “Please, change “…of estrogen” to “… of estrogens”.
  6. Line 41-43: “One of the most… anti-breast cancer activity.”. Please, move this sentence at the end of the paragraph. Combine and modify the text accordingly.
  7. Line 43: “The coumarin…” Please, start a new paragraph at this point.
  8. Line 50: Please, change “…its SERMs activity…” to “…its SERM activity…”.
  9. Line 52: Please, add “…steroid derivatives”.
  10. Line 53: Please, change “…Anti-breast…” to “…anti-breast…”.
  11. Line 56-8: “These compounds…. uterine cytosol.”. Please, provide representative references.
  12. Line 67-9 “…while keeping methoxy group at the aromatic ring.…relative to one another.” Τhis sentence is not clear and should be rephrased.
  13. Scheme 1: Compound Ib and 1b are identical. The scheme must be corrected/redesigned.
  14. Line 86-91 and 97-117. In these lines the authors describe the characteristic IR and NMR peaks for the synthesized compounds.

To my opinion, this part of the manuscript should be limited as the majority of the described signals are obvious. Reference has to be made only to important diagnostic peaks that confirm the corresponding structures.

  1. Lines 120-133: The biological evaluation of intermediates Ia-b and IIa-b should be described first, as it is noted in the text: “The biological investigation was started…and IIa-b (Table 1).” Consequently, this part of the mansucript should be reorganized.
  2. Line 151: I consider that most of the compounds were not highly cytotoxic, but more cytotoxic than camptothecin. I would characterize only compounds IIIe,c, f as highly cytotoxic.
  3. Line 166-168: Please, change “Compound Ib showed a significant inhibition of the aromatase enzyme with IC50 value of 14.5 μM..” to “Compound Ib showed equipotent to letrozole (IC50 = 15.03 μM) aromatase inhibitory activity with IC50 value of 14.5 μM (Table 4 and Figure 5).
  4. Line 202-3: Please, correct “2-(7-methoxy-2-oxo-2H-chromen-4-yl)acetic acid”.
  5. Line 227: Please, change “thirty five minutes” to “35 min”.
  6. Line 229: Please, recalculate the number of mmoles.
  7. Line 243, 251: Please, recalculate the yields.

In general, the manuscript is interesting but it needs a careful revision by the authors.  On this basis, I would recommend publication only after minor revision.

Author Response

Please see the attavhment
